# OpenReview forum: "Reasoning Theater: Disentangling Model Beliefs from Chain-of-Thought"
_ICML.cc/2026/Conference — ICML 2026 regular_

### Official Review · Reviewer_RwHc · 2026-03-11

**Soundness:** 3
**Presentation:** 3
**Significance:** 3
**Originality:** 2
**Overall Recommendation:** 4
**Confidence:** 3

**Summary:**

Reasoning models performance chain-of-thought (CoT) before outputting their final answer. This paper uses two methodologies to assess whether a reasoning model has "decided" on its final answer ahead of time, during its CoT. The two methodologies are attention probes on activations, and early forced answering; these methodologies are both roughly equally as effective. Using these methodologies in addition to an external LLM judge, one can identify "performative reasoning" - cases where the model "knows" what its final answer will be, but reasons as if it does not. The authors find that performative reasoning is more frequent in recall-based benchmarks, and less frequent in multi-step reasoning benchmarks.

**Compliance With Llm Reviewing Policy:**

Affirmed.

**Final Justification:**

In my view, the main limitations of this work are that it is limited to multiple choice settings, and that it is evaluated on a small number of benchmarks (MMLU, GPQA). I think these limitations should have been discussed more explicitly in the manuscript; for example, I still don't understand whether being restricted to the multiple choice setting is a fundamental limitation here (I think it is).

Overall, I think the paper, within its domain of multiple choice evaluation, is interesting and valuable. I therefore maintain my positive score of 4: Weak accept.

**Key Questions For Authors:**

- How many total activations are used for probe training? How does probe performance scale with training dataset size?
- Is "final answer" the proper signal to train attention probes to predict?
  - An alternative is to train attention probes to predict the early forced answer prediction, which would make it so that attention probes predict "what answer the model currently thinks is correct". Have you considered this? Is one methodology more principled than the other?

**Limitations:**

The authors don't discuss limitations. I would encourage the authors to add explicit discussion of limitations; including the fact that the study is limited to two multiple choice benchmarks, and that it is unclear how to extend the methodology to non-multiple-choice settings.

**Strengths And Weaknesses:**

**Strengths**

- Important domain
  - Reasoning models have become pervasive over the past year because of the jump in capabilities that they've enabled. Studying the properties of these reasoning models is valuable, and developing tools to understand them is important.
- Thorough analysis and baselines
  - The authors conduct diligent baselines and ablations in the appendix, including comparison to vanilla (non-attention) probes, and a random label baseline.
- Well written
  - The paper is clear and easy to read.

**Weaknesses**

- Limited evaluation domains
  - The authors evaluate on two benchmarks: MMLU-Redux and GPQA-Diamond. I would be curious to see more thorough evaluations across more benchmarks, including other domains like mathematics.
  - Additionally, the study is limited to multiple choice evaluations; it is not clear how to go beyond multiple choice.

---

> ### Author Rebuttal · Authors · 2026-03-31
>
> We thank the reviewer for their thoughtful comments and questions.
>
> * While we were unable to add additional benchmarks to our analysis at this time due to the size of our original models and activation sets, we would like to note that MMLU *does* include domains such as mathematics (https://arxiv.org/abs/2009.03300). When we compare probe accuracy on specific MMLU domains, we again see a similar trend in performative reasoning occurring on less difficult tasks that require only recall like  “High School World History,” “Virology,” and “High School Biology” while the smallest gaps were in multi-step reasoning domains like “High School Mathematics,” “Abstract Algebra,” and “College Physics” (Table 2 in rebuttal to Reviewer jKNA). We also added additional analysis with four new models, specifically the DeepSeek R1 Qwen Distills (1.5B, 7B, 14B, 32B) in Table 1 of rebuttal to Reviewer jKNA. As model size decreases, MMLU becomes harder for the model, and the best probes require more of a reasoning trace to predict the final answer compared to the CoT monitor. This confirms our hypothesis about task difficulty leading to performative reasoning along another axis (model size).
>
> * While our current methodology is limited to multiple choice evaluations, we felt that this was enough predictive power to answer the main research questions of our work which focus on whether reasoning models’ CoT is performative, and which characteristics of a dataset lead to that through the study of an outcome distribution.
>
> **Questions**
> 1. We use a total of 6,488,230 token activations across 4170 questions (80% of MMLU) for attention probe training. We show the results of probe performance with training dataset size below, finding that performance plateaus at around 1667 questions in Table 4 below.
> 2. One issue with relying on early forced answer predictions as a training signal is that it sets an upper bound on the performance of the attention probe. We find that in GPT-OSS, probes approach and can even exceed forced answering in some cases, indicating an inability for the model to faithfully predict its final answer in text. In addition, we find that probes regularly exceed forced answering when studying distilled models, further indicating that models are not always able to explain their own internals faithfully (Table 5 below). For this reason, we train to the objective of "final answer."
>
>
> **Table 4: Probe vs. Forced Answer Overall Accuracy (MMLU)**
>
> | Model | Best Layer | Probe Acc. | Forced Acc. | Probe − Forced (pp) |
> |:------|----------:|-----------:|------------:|--------------------:|
> | R1-Distill-Qwen-1.5B | 18 | 67.1% | 21.6% | +45.5 |
> | R1-Distill-Qwen-7B | 18 | 65.2% | 36.9% | +28.3 |
> | R1-Distill-Qwen-14B | 46 | 77.5% | 66.9% | +10.6 |
> | R1-Distill-Qwen-32B | 62 | 76.3% | 57.6% | +18.7 |
> | DeepSeek-R1-671B | 36 | 86.8% | 93.0% | -6.2 |
>
>
> We show the average accuracy for probes and forced answering, averaged across all reasoning steps, measured against the model's final answer. The probe consistently outperforms forced answering for distilled models, where the gap widens as model size decreases: the 1.5B and 7B models fail to verbalize commitments that the probe can already read from hidden states. The 671B model is the exception: forced answering slightly outperforms the probe, consistent with the full-size model being capable enough to articulate its intermediate beliefs when prompted.
>
> **Table 5: Probe Accuracy vs. Training Set Size (DeepSeek-R1-671B, Layer 36, MMLU)**
> | Training Questions | Token Activations | Test Accuracy |
> |-------------------:|------------------:|--------------:|
> | 417 | 636,160 | 23.0% |
> | 833 | 1,297,855 | 48.4% |
> | 1,250 | 1,921,935 | 67.1% |
> | 1,667 | 2,556,804 | 86.1% |
> | 2,085 | 3,230,782 | 86.5% |
> | 2,502 | 3,922,329 | 86.5% |
> | 2,919 | 4,550,184 | 87.5% |
> | 3,336 | 5,173,970 | 86.4% |
> | 3,753 | 5,838,062 | 86.3% |
> | 4,170 | 6,488,230 | 87.6% |
>
> We find that probe accuracy saturates at \~1,667 training questions (~2.5M token activations), with no further gains from increasing the dataset past that. Val/test sets are fixed at the same held out 522 questions each. All runs use identical hyperparameters (lr=1e-3, wd=1e-3, 20 epochs).
>
> [1] Bigelow et al. 2025., Forking Paths in Neural Text Generation.

---

> > ### Author Rebuttal · Reviewer_RwHc · 2026-04-02
> >
> > I thank the authors for their detailed response.
> >
> > I think the multiple choice setting is indeed a good one to study this question scientifically, but I think being limited to the multiple choice setting precludes the work from"[establishing] attention probing as an efficient method for detecting performative reasoning and for adaptive computation in reasoning LLMs" generally, as claimed in the abstract.
> >
> > I also am not satisfactorily convinced that training a probe to predict final output rather than forced output is the principled thing to do, and I think this deserves more careful thought and discussion.
> >
> > That said, I think this is an interesting and valuable paper, and maintain my weak acceptance rating.

---

> > > ### Author Response · Authors · 2026-04-06
> > >
> > > We thank the reviewer for their comments.
> > >
> > > We agree that training a probe to predict the model’s forced output could be an interesting additional experiment. That said, we chose to train on the model’s final answer for two main reasons. First, forced answering works poorly for smaller models in our experiments, serving as a lossy representation of “what the model currently thinks is correct.” As we show with the distilled Qwen models (particularly the 1.5B model), we can often probe out answer information substantially earlier than we can force the model to emit that answer in text. This indicates that the model’s internal belief is easily extractable but cannot be elicited from prompting, and training to the model’s forced answer may underestimate what is already present in the model’s internal state.
> > >
> > > Second, if a model is being unfaithful in its chain-of-thought, its forced answer may also be unfaithful or incomplete as a behavioral readout. Recent work has shown that training models on activations can reveal hidden beliefs which are not elicited from prompting directly [2,3]. In these cases where the forced and final answer diverge, training a probe to predict forced answers would risk inheriting the same failure mode, whereas predicting the model’s eventual final answer is a closer approximation to the model’s committed behavior.
> > >
> > > For that reason, we view final-answer prediction as the more appropriate target for the question we study here: when information about the model’s eventual decision becomes decodable from its internal activations.
> > >
> > > [2] Karvonen, Adam, et al. "Activation oracles: Training and evaluating llms as general-purpose activation explainers." arXiv preprint arXiv:2512.15674 (2025).
> > >
> > > [3] Huang, Vincent, et al. "Predictive concept decoders: Training scalable end-to-end interpretability assistants." arXiv preprint arXiv:2512.15712 (2025).

---

### Official Review · Reviewer_zaPT · 2026-03-11

**Soundness:** 3
**Presentation:** 3
**Significance:** 3
**Originality:** 3
**Overall Recommendation:** 4
**Confidence:** 5

**Summary:**

This work focuses on whether the chain-of-thought (CoT) explanations produced by large language models faithfully represent their internal reasoning computation. To investigate this, the authors employ attention probes, forced answering, and CoT monitoring to track when models commit to a final answer across different datasets. They discover that on simpler, recall-based tasks, models often know the answer internally long before revealing it in their text, whereas on complex tasks, internal beliefs and textual reasoning evolve in tandem. Furthermore, the researchers demonstrate that calibrated attention probes can be used for early exiting, significantly reducing token generation costs without sacrificing accuracy. The core reasearch problem is "performative reasoning," which carries significant practical implications for LLM safety, interpretability, and adaptive inference efficiency.

**Compliance With Llm Reviewing Policy:**

Affirmed.

**Final Justification:**

Most of my concerns have been addressed. See rebuttle acknowledgement.

**Key Questions For Authors:**

The tested two benchmarks focus on memory and reasoning, respectively. So, is it possible that the difference is task type rather than your clamied difficulty? What if we compare the accuracy gap between thinking model and no-thinking mode? For example, memory-focused tasks may witness smaller gap.

**Limitations:**

yes

**Strengths And Weaknesses:**

**Strength:**

1. The paper translates the issue of model "faithfulness" into a temporal misalignment between internal predictions and external text outputs. It introduces the concept of performative CoT, which directly addresses a significant pain point in current RL-based large language models.

2. The study comprehensively combines white-box feature extraction (attention probes), gray-box behavioral intervention (forced answering via early truncation), and black-box text evaluation (third-party LLM monitoring). The high consistency of the data obtained across these three distinct methods makes the conclusions highly convincing.

3. The paper proposes and validates a probe-based "Early Exit" mechanism. By saving up to 80% of generated tokens while maintaining accuracy, this approach offers direct, practical reference value for AI developers looking to significantly reduce inference costs.

**Weakness:**
1. The research and the training of the probe classifiers rely on only two tasks, raising potential concern about generalization and task bias. Meanwhile, in real-world mathematical proofs, open-ended code generation, or pure generative reasoning tasks where the answer is not a simple A/B/C/D classification, this probing method would be difficult to apply directly.

2. Insufficient exploration of root causes. While the authors successfully observed the phenomenon, they did not perform ablation studies on possible causes. For instance, is this "performative reasoning" a side effect of hacking long-length rewards during RL training? Analysis beyond phenomenological observation could deeper this research.

---

> ### Author Rebuttal · Authors · 2026-03-31
>
> We thank the reviewer for the positive assessment and helpful suggestions, and address their comments and questions below.
>
> **Responses to Weaknesses:**
> 1. We have added additional analysis with four new models, specifically the DeepSeek R1 Qwen Distills (1.5B, 7B, 14B, 32B), see Table 1 in rebuttal to reviewer jKNA for results. As model size decreases, MMLU becomes harder for the model, and the best probes require more of a reasoning trace to predict the final answer compared to the CoT monitor. This confirms our hypothesis about task difficulty leading to performative reasoning along another axis (model size). Additionally, we break down MMLU into its 57 subsets and find the same trend as well. We find that the MMLU domains with the largest probe minus CoT monitor gap were easier due to recall, while the smallest gaps were in multi-step reasoning domains (Table 2 in rebuttal to reviewer jKNA).
>
> We would also like to note that we do see evidence of probe generalization, as shown by probes generalizing from MMLU to GPQA-Diamond via direct transfer. While we are unable to apply probes currently to non-classification tasks, we find that this approach is enough to study performative reasoning by comparing to black box methods, which is the focus of our work.
>
> 2. Unfortunately, we do not have access to checkpoints during RL training but do believe that this is largely due to using long-length CoTs with little penalty. This is characterized in the DeepSeekR1 technical report [1, Figure 1], where rollouts grow longer at the end of RL training while accuracy stagnates. While we did not focus our analysis on why performative reasoning appears, we have added further analysis in documenting whether verbalized inflection points occur due to internal belief shifts or if they are performative (Table 3 in rebuttal to reviewer jKNA). We find weak evidence for inflection points occurring within a window of probe confidence shifts in MMLU, but do not see a clear causal relationship between the two.
>
> [1] https://arxiv.org/abs/2501.12948
>
> **Questions**
> We have included further analysis to reinforce our claim about task difficulty. For example, when holding the task type constant but looking at different model sizes, we again see the same trend: smaller models (which find the task harder) are less performative. When we look at different domains in MMLU, we see multi-step reasoning questions in math and physics being more performative than memorization-based history and geography. Our intuition is that thinking models are significantly more performative than non-thinking models, due to their SFT and RL stages where they are prompted to think step-by-step. This leads to pressure to produce longer responses with phenomena like backtracking and verification, even when models may have high internal confidence within an answer.
>
> While we do not do any analysis on non-thinking models, we agree this analysis would be interesting to study, especially over the course of post-training and will acknowledge this as a potential future direction!

---

> > ### Author Rebuttal · Reviewer_zaPT · 2026-04-03
> >
> > Thank the authors for their response.
> >
> > W1: Partially resolved. I am not convinced that two multiple-choice benchmarks fully exhibit generalization. Since this paper targets LLM CoT, merely testing on multiple-choice tasks could be a potential limitation.
> >
> > W2: Addressed.
> >
> > KQ: Addressed.
> >
> > Overall, I will keep this possitive score.

---

> > > ### Author Response · Authors · 2026-04-06
> > >
> > > We thank the reviewer for their response. While we are unable to evaluate our models on other benchmarks at this time, we note that on GPQA, probes nearly match forced answering performance despite using unseen data and in some cases outperform it. Although we do not probe for non-multiple-choice tasks, we find that this suffices to answer our central question of *whether models performatively reason.* We believe that this methodology can still be utilized for many safety benchmarks as they are based on binary evaluations.

---

### Official Review · Reviewer_jKNA · 2026-03-15

**Soundness:** 3
**Presentation:** 3
**Significance:** 3
**Originality:** 3
**Overall Recommendation:** 5
**Confidence:** 3

**Summary:**

This study investigates whether the CoT of the LLM correctly reflects the actual internal state of the LLM. In particular, we conducted an analytical survey using attention probe from the viewpoint of interpretability and internal mechanism that the LLM outputes CoT unnecessarily according to the internal pressure for CoT, regardless of whether the LLM has reached a conclusion internally at an early stage. The target language models are DeepSee-R1 and GPT-OSS, which are famous as the Reasoning Model, and analyzes the state of GPQA and MMLU tasks when performing GPQA and MMLU tasks. In addition, as a bonus, it is proposed to implement the early termination of Reasoning based on the analysis results.

**Compliance With Llm Reviewing Policy:**

Affirmed.

**Final Justification:**

I thank the authors. I acknowledge the novelty and performance of this method. I update each score. it is worth to be discussed in the conference.

**Key Questions For Authors:**

Both DeepSeek R1 and GPT-OSS-120B are MoE models, and Parameter that is actually activated should be limited, and I think that consideration is necessary for that in the attache probe, consideration is necessary for that, but there were any points or concerns that you devised in that regard? If not, can you tell me why you don’t need it?

**Limitations:**

yes

**Strengths And Weaknesses:**

Strengths:
- Soundness: We are comprehensively investigating items that need to be examined, no problem


Weaknesses/Comments:
- Significance: The method of analyzing changes in the Attenstion Probe and Confidence during the CoT itself is in favor of it, with no praise or criticism in an orthodox way. The adoption of DeepSeekR1 and GPT OSS 120B in the target language model is also orthodox and without praise or criticism. Reasoning The ability to return the output early in the middle of it is in itself, without praise, criticism, and I agree with it. Overall, it is a solid collection of fact-checking and evidence-gazing, and although I like it, it is not sigificant.

- Presentation: Figure1 is not a diagram that points to the whole picture of a project that can only be explained by a diagram, but rather the contents of the Introduction and Caption are very easy to understand, and the intention of the figure is difficult to understand. Figure2 and Figure4 depict the ideas of this research, and if you use this as a reference to Figure1, it seems that the reader can quickly understand the intention.

- Originality: Similar to Siginificance, it is not a proposal of a new method, but a solid survey using existing methods, and I do not think that it matches the Originality as required by ICML. I look forward to the reports at workshops.

---

> ### Author Rebuttal · Authors · 2026-03-31
>
> We thank the reviewer for their thoughtful comments and questions, which we address below.
>
> **Significance**
> * The main contribution of our work is that CoT appears to be performative in certain settings, and that we can use the relative confidence of methods like probes to evaluate this and compare to black box monitors. We have included additional analysis on four other models of varying sizes to show performativity trends based on pre-trained knowledge. In addition, we compare probe confidence shifts at the step level to verbalized inflection points to see if internal belief shifts lead to these inflections. Our work is the first to directly compare black box and white box methods on model internal confidence across the sequence dimension, and we show this phenomena exists based on task difficulty, which is novel.
>
> **Presentation**
> * Our hope was to summarize our three methods of decoding the final answer in Figure 1 without overwhelming the reader with our results as well. We have since updated the paper to show Figure 3 earlier, clarifying the central takeaways from the paper to readers.
>
> **Originality**
> * While the use of probes for predicting the final answer is not novel, our methodology for training this probe at the token level to predict the outcome distribution at any prefix is novel. This allows us to do direct confidence-based early exit without reading CoT, as well as characterizing the difference between text and activations in terms of model belief updates.
>
> Response to Question:
> 1. We do not need to consider the MoE expert layers in our setup as the probes are trained on the entire layer’s output activation, not specifically the MLP layers. Any information from the MoE layers ends up in these activations as well.
>
> **_We now include tables referenced in other rebuttals strengthening our claims with additional experiments._**
>
> **Table 1: Comparing Probe Accuracy of Final Answer for Qwen Distill Series (MMLU)**
>
> *Note: The probe accuracy is using the objective of predicting the model's final answer, and not the correct answer.*
>
> | Model | Model Orig. Acc. | Best Layer | Probe @ 0% | Probe @ 25% | Probe @ 50% | Probe @ 75% | Probe @ 100% |
> |:------|-----------------:|----------:|-----------:|------------:|------------:|------------:|--------------:|
> | R1-Distill-Qwen-1.5B | 51.2% | 18 | 46.6% | 51.7% | 64.6% | 87.8% | 95.6% |
> | R1-Distill-Qwen-7B | 67.4% | 18 | 63.8% | 68.2% | 72.1% | 77.8% | 82.0% |
> | R1-Distill-Qwen-14B | 84.3% | 46 | 70.2% | 72.7% | 81.0% | 87.0% | 97.6% |
> | R1-Distill-Qwen-32B | 88.8% | 62 | 77.2% | 75.6% | 83.3% | 88.6% | 96.1% |
> | DeepSeek-R1-671B | 92.5% | 36 | 81.3% | 80.0% | 90.1% | 93.8% | 99.1% |
>
> Larger models achieve higher probe accuracy earlier in reasoning, likely reflecting greater in-weights knowledge. The 671B probe reaches high accuracy quickly and plateaus, the distilled models (7B–32B) remain flat before rising late in the sequence, and the 1.5B probe sharply increases only in the second half of reasoning.
>
>
> **Table 2: Probe vs. CoT Monitor Gap Across MMLU Domains (at 25% through CoT)**
> | Domain | n | Probe Acc. | CoT Monitor Acc. | Gap (pp) |
> |:-------|--:|-----------:|-----------------:|---------:|
> | Us Foreign Policy | 44 | 95.5% | 30.1% | +65.3 |
> | High School Geography | 51 | 94.1% | 31.4% | +62.7 |
> | Virology | 27 | 92.6% | 32.4% | +60.2 |
> | High School Government And Politics | 64 | 93.8% | 37.9% | +55.9 |
> | High School World History | 64 | 96.9% | 41.4% | +55.5 |
> | ... | | | | |
> | College Mathematics | 44 | 68.2% | 74.4% | -6.2 |
> | High School Mathematics | 43 | 67.4% | 68.6% | -1.2 |
> | Abstract Algebra | 55 | 69.1% | 66.8% | +2.3 |
> | Moral Scenarios | 61 | 68.9% | 65.6% | +3.3 |
> | College Physics | 53 | 69.8% | 64.6% | +5.2 |
> | **Overall** | **2930** | **84.9%** | **47.6%** | **+37.3** |
>
> We show that the same task difficulty dependent trends exist within subsets of MMLU. The most performative CoTs occur for recall/memorization based domains such as Geography, World History, and Foreign Policy. Meanwhile, the least performative CoTs occur for multi-step reasoning questions such as Mathematics, Physics, and Moral Scenarios (where models reason back and forth for different sides in CoT).
>
> **Table 3**
> | Model / Dataset | # Inflections | P(inf∈window ∣ shift) | P(inf∈window ∣ no shift) | Δ (pp) |
> |:----------------|-------------:|---------------------:|-------------------------:|-------:|
> | R1 / MMLU | 805 | 43.1% | 27.2% | +16.0 |
> | R1 / GPQA | 946 | 18.6% | 20.1% | -1.5 |
> | GPT-OSS / MMLU | 741 | 53.8% | 42.4% | +11.4 |
> | GPT-OSS / GPQA | 550 | 29.6% | 57.5% | -27.9 |
>
>
> We conduct additional analysis on inflection points, finding that for MMLU inflections occur more often within a window of a probe confidence shift. We conclude that there is not a clear causal relationship between individual inflection points and shifts in internal confidence, but rather a response level correlation between inflections and overall confidence across the sequence.

---

> > ### Author Rebuttal · Reviewer_jKNA · 2026-04-06
> >
> > The results of the experiment in the rebuttal worked in the direction of strengthening my confidence. This research has been well-researched, and I think that including these in the paper will make it even more useful for readers.
> >
> > Novelty and significance should be considered separately, and require a more detailed rebuttal of the Significance (how this study promotes research into language models).
> > The opening rebuttal does not support originality or significance.
> >
> >
> > As for the presentation and originality, I think it was something that satisfied me. Therefore, although the score was partially increased, it did not reach a comprehensive evaluation.

---

> > > ### Author Response · Authors · 2026-04-07
> > >
> > > We thank the reviewer for their continued engagement and positive feedback. We would like to respectfully seek clarification: the post-rebuttal comment suggests the significance rebuttal was insufficient, but does not identify which specific claims remain unaddressed. We hope the points below help resolve this. Moreover, it was unclear which parts of the paper the reviewer found satisfactorily and unsatisfactorily original. We argue the significance of this work rests on three concrete contributions to our understanding of performative reasoning:
> > >
> > > 1. *Difficulty-dependent performativity is a novel and actionable finding.* Prior work has documented CoT unfaithfulness, but our results are the first to show a systematic, measurable split based on task difficulty: MMLU exhibiting strong performativity while GPQA exhibits near-zero. This tells researchers precisely when to expect performative CoT, which prior work could not characterize.
> > > 2. *The attention probe methodology enables a new mode of analysis.* Training probes at the granular token level to track belief state evolution throughout generation, rather than at a single point, is a novel methodological contribution. This is what allows us to quantify performativity at all, and we hope it serves as a reusable tool for future faithfulness research, particularly in tandem with forced answering and monitoring as we did in this work. Finally, we hope others can leverage similar probes to enable confidence-based early exit, given that we showed that our probes save up to 80% of tokens on MMLU and 30% of tokens on GPQA while maintaining similar accuracy.
> > > 3. *Inflection points are shown to reflect genuine belief updates.* The finding that backtracking and 'aha' moments are strongly concentrated in non-high-confidence traces provides the first quantitative evidence that these commonly-studied reasoning steps are meaningful signals rather than stylistic artifacts.
> > >
> > > We hope this more directly addresses the significance concern, and welcome any further specific feedback.

---

### Official Review · Reviewer_g9zx · 2026-03-16

**Soundness:** 3
**Presentation:** 2
**Significance:** 2
**Originality:** 2
**Overall Recommendation:** 4
**Confidence:** 3

**Summary:**

This paper compares three methods for decoding the LLM’s final answer from partial CoTs, and reveals the phenomenon of performative CoT, where the internal representations of the model show strong confidence in the final answer, but the model continues generating excess tokens before producing the answer. Specifically, the three methods are attention probes that are trained to predict the answers based on internal representations, forced answer that inserts a prompt after partial CoTs to ask for an answer, and CoT monitor that reads answers from the partial CoTs, respectively. It is observed that both DeepSeek-R1 and GPT-OSS exhibit high probe accuracy at the beginning of CoT traces on MMLU, while their probe accuracy gradually increase from low to high as the reasoning traces expand on GPQA. Compared to the attention probe, the CoT monitor starts from a random-guessing accuracy on both datasets, and its accuracy increases more slowly. The authors also observe that most inflection points (e.g. aha moments) are genuine in the sense they change the prediction of the attention probe.

**Compliance With Llm Reviewing Policy:**

Affirmed.

**Final Justification:**

The authors clarified that the experiments are conducted with separate train/valid/test datasets, which addresses my major concern on the soundness of this paper. Therefore, I decide to increase my rating accordingly.

**Key Questions For Authors:**

1. The term “attention probes” is ambiguous, as it may refer to probes that take the attention matrices of Transformers as input. Better to call it “attentional probes”.
2. While the authors argue that linear probes don’t work, [1] uses a linear probe with conformal prediction. Since simpler probes are better for interpretability, why don’t the authors consider a similar approach?
3. Do you generate train & valid splits for training the attention probe? If not, the probe might just learn to memorize the mapping from partial CoTs to the final answers.
4. What’s the purpose of Table 1 w.r.t. the contribution of this paper?
5. Is the probe on GPT-OSS (GPQA) close to random guess? If we look at P=1, 0.370 + 0.124 = 0.494. If the probe performs random guessing, later experiments become meaningless.
6. Line 266 - 269 right: Why do you use quadratic fits? Why do you measure the difference in slopes?
7. Figure 3: Does the y axis accuracy refer to whether the probe prediction equals to the final answer, not necessarily the ground truth? Then probe accuracy just suggests the compressibility. Otherwise, I can’t understand why the maximum of y axis can be 1.0.
8. Line 365: Can you explain what the calibration is here?

**Limitations:**

No limitations discussed. The authors are suggested to discuss at least the scope of their method, such as on what tasks and domains their method will likely work.

**Strengths And Weaknesses:**

Soundness:

- Weaknesses:
    - This paper assumes that a high confidence from the probe at the beginning of a CoT indicates the rest of the CoT is performative, which is invalid in my opinion. Since the probe is trained to predict the final answer of a partial CoT on a dataset, it serves as post-hoc compression of the CoTs. My understanding is that a high confidence only suggests the rest of CoTs are compressible when you know the model’s past behaviors on this dataset. It doesn’t necessarily mean the rest of CoT is easy enough to decode, especially given the probe is not linear.
    - This paper assumes that a higher accuracy of forced answer (note by accuracy, the authors mean whether it equals to the final answer generated in the CoT, not the ground truth) than CoT monitor indicates the rest of the CoT is performative. This is also somehow problematic. In my opinion, this can only demonstrate the model doesn’t stop generating tokens before achieving very high accuracy. It doesn’t necessarily mean the rest of CoT is performative.
    - The authors only investigate two datasets, MMLU and GPQA, but claims that the accuracy of attention probes depends on the difficulty of the dataset. This is overclaim, as there are many controlling factors between these two datasets. The authors need more datasets to support this claim.

Presentation:

- Weaknesses:
    - The title has little to do with what is studied in this paper. My understanding of disentanglement is that one can separately control the computation needed for CoT generation, and the length of CoTs.
    - The idea of training probes over internal representations to perform early-stop over CoTs have already been studied in previous papers [1, 2]. The authors didn’t cite them nor discuss their distinct contributions.

Significance:

- Weaknesses:
    - Given that CoT early stopping has been studied in [1, 2] and no comparison in this paper, it is not clear to me whether the token savings achieved in this paper is significant. It might be possible that post-hoc compression of CoT is easy, since the model is general and isn’t optimized for token savings on a single dataset.
    - Two datasets and two models are on the weak side of experimentation for LLM papers.

Originality:

- Weaknesses: Very limited given the understanding already demonstrated in [1, 2].

[1] Ashok and May. Language Models Can Predict Their Own Behavior. NeurIPS 2025.

[2] Afzal et al. Knowing Before Saying: LLM Representations Encode Information About Chain-of-Thought Success Before Completion. ACL Findings 2025.

---

> ### Author Rebuttal · Authors · 2026-03-31
>
> We thank the reviewer for their thoughtful comments and questions, which we address one-by-one below. We reference tables included in rebuttal to jKNA due to char. limits.
>
> **Soundness**
> * We clarify that our definition of performative CoT is based on the relative gap between the probe and CoT monitor. The ability to probe final answer information directly from the residual stream across the sequence dimension indicates that this is not just post-hoc compression of CoT. We note that attention probes, although not linear, are essentially a learned weighted average of token-level linear predictions.
> * We use *performative* in an operational sense: the model can be induced to reveal its eventual answer earlier than expressed in CoT, and thus additional tokens generated were not necessary. The model continuing to generate tokens despite having high accuracy is what we label performative, since it “knew” its final answer already.
> * We’ve added two new sets of analyses to strengthen our claim about task difficulty (recall-based vs multi-step reasoning tasks). 1) We ran the same analysis on the DeepSeek-R1 Qwen distilled models in Table 1. As model size decreases, MMLU becomes harder for the model, and the best probes require a longer portion of a reasoning trace to predict the final answer. 2) We include further analysis on subsets of MMLU, looking at different domains within the dataset (Table 2). The MMLU domains with the largest (probe - CoT monitor) gap were easier due to recall, while the smallest gaps were in multi-step reasoning domains.
>
> **Presentation**
> * We have updated our title to include “Disentangling Model Beliefs from CoT” so as to not make claims about model internal computation.
> * We apologize for the oversight and will include both works in the revision! While [1,2] uses probes for early prediction, our focus differs: we train probes that predict the model’s *final answer distribution at arbitrary prefixes with token-level granularity*. This enables us to study the *dynamics of how internal answer commitment evolves over reasoning*, and particularly the mismatch between internal confidence and verbalized CoT, which is not explored in prior work.
>
> **Significance**
> * Although early stopping has been studied in [1,2], the probes used in those papers do not enable adaptive inference at any token of generation conditioned on confidence. We study reasoning model CoTs (on the order of 1-10k tokens compared to ~300 in [1,2]). [1] only early stops at the end of the prompt, and [2] does early stopping at set intermediate points while also requiring an additional forced answer prompt.
>
> **Originality**
> * The main focus here is not simply that probes can predict future outcomes early, but that they allow us to study *performative CoT*: cases where internal answer belief and expressed reasoning diverge over the course of a long trace. We believe this focus on the *dynamics* between internal belief and verbalized reasoning, especially in frontier open-weight long-trace reasoning models, is distinct from [1,2].
>
> Question Responses:
> 1. We followed terminology from previous work (Kantamneni et al. 2025 A Case Study in Sparse Probing) to refer to the probes as attention probes, and will clarify this architecture from probes trained on attention outputs in the next iteration of the paper.
>
> 2. The reason we do not adopt a [1]-style linear probe is that conformal prediction is about calibrating confidence once you already have a useful score/output. To our knowledge, it does not fix a probe that is failing to read out the target in the first place.
>
> 3. Yes, we generate train/val/test set splits that contain disjoint sets of questions from the dataset. There are no overlapping questions between sets, so it is not possible to memorize partial CoTs to the final answer.
>
> 4. We have since removed this table from our paper’s main body as it is not directly relevant to the main questions we are trying to answer.
>
> 5. For our multiple choice tasks, random guess would be ~25% accuracy as there are four choices. 0.494 is still well-above random chance for GPT-OSS on GPQA.
>
> 6. We use quadratic fits to smooth over noise which arises from binning our predictions into different segments of the rollout. We measure the difference in slopes because we are interested in the incremental difference between our probes/forced answering and the CoT monitor over a single step for performative reasoning.
>
> 7. The y-axis accuracy refers to if the prediction equals the final (not ground truth) answer. Since the probe accuracy is significantly higher than the CoT monitor for the same prefix, it cannot be compressing the literal text and must be leveraging some signal from the internals that is not available to the CoT monitor.
>
> 8. Calibration measures the correlation between the model’s confidence and its prediction success rate given that confidence. Ex: if the model is correct 20% of the time when it is 20% confident, that’s well-calibrated.

---

> > ### Author Rebuttal · Reviewer_g9zx · 2026-04-04
> >
> > Thanks the authors for clarifying their definition of performative. Now it's clear to me that a higher accuracy of forced answer suggests the rest of CoT is performative. However, it remains a concern that the probe is trained and evaluated on the same set of reasoning traces, which may serve as a post-hoc compression. Only when the probe is evaluated on a set of problems and reasoning traces other than the one it is trained on, we can say the final answer can be predicted earlier from the internal belief of the model. If the authors can show this, I am happy to increase my rating.

---

> > > ### Author Response · Authors · 2026-04-06
> > >
> > > We thank the reviewer for their response.
> > >
> > > Regarding the remaining concern, we would like to re-iterate our clarification that *the probe is not trained and evaluated on the same set of reasoning traces*. Rather, the set of MMLU reasoning traces are separated into a train, validation, and test set (these are disjoint sets of questions). In addition, probes perform similarly to forced answering even when being evaluated on an entirely different dataset (GPQA-Diamond).

---

### Decision · Program_Chairs · 2026-04-30

**Decision:**

Accept (regular)

**Comment:**

The reviewers praised the paper for studying the important problem of CoT faithfulness, they raised many issues regarding the narrow scope of the evaluation on MCQ tasks, the methodology of the probe training, the lack of originality, and the broad claims made in the title / abstract / intro of the paper which are generally not validated by the rest of the paper.

Reviewers praised the paper for a consistent evaluation of early stopping potential in CoT models with attention probes, forced early answering via truncation, and 3p monitoring by another LLM, and the use of baselines produces a good evaluation. The problem was recognized as an important topic.

While the reviewers shared many concerns with the paper, many of these concerns seems to have been addressed in the review period with two reviewers increasing their scores. The AC believes many concerns have not been addressed. However, given that all reviewers recommend acceptance and two have updated their scores positively suggesting that at least some concerns have been alleviated I as SAC recommend acceptance.

That said there are outstanding reviewer concerns that I do think merit careful response and revision by the authors as noted by the AC:

Reviewer g9zx had issues with the setup and concept of performative CoT, the presentation -in particular the title and the lack of proper reference to prior work- and did not feel that the experimental evaluation was sufficiently comprehensive because it only includes 2 models and 2 datasets. In general, the reviewer did not feel that the paper had significant new findings. The authors clarified their definition of performative CoT, after which Reviewer g9zx increased their rating to a weak accept. Issues on significance, breadth of evaluation, and references to prior work remain outstanding.

Reviewer jKNA shared concerns with the issues on significance and felt that the paper was more appropriate as a workshop submission. The authors provided more experimental results in the rebuttal, but the reviewer still felt that the lack of significance was an outstanding issue.

Reviewer zaPT was concerned that the authors don't provide any explanation of why performative CoT may emerge, and again had issues with the evaluations only being done on 2 datasets. After the rebuttal, their concerns remained outstanding.

Reviewer RwHc again raised the issue of only evaluation on 2 multiple-choice datasets.